# [Re] Classwise-Shapley values for data valuation

**Markus Semmler**                                                  *m.semmler@appliedai.de*
*appliedAI Initiative GmbH*

**Miguel de Benito Delgado**                                        *miguel@appliedai.de*
*appliedAI Institute gGmbH*

**Reviewed on OpenReview:** *https://openreview.net/forum?id=srFEYJkqD7*

## Abstract

We evaluate CS-Shapley, a data valuation method introduced in Schoch et al. (2022) for classification problems. We repeat the experiments in the paper, including two additional methods, the Least Core (Yan & Procaccia, 2021) and Data Banzhaf (Wang & Jia, 2023), a comparison not found in the literature. We include more conservative error estimates and additional metrics, like rank stability, and a variance-corrected version of Weighted Accuracy Drop, originally introduced in Schoch et al. (2022). We conclude that while CS-Shapley helps in the scenarios it was originally tested in, in particular for the detection of corrupted labels, it is outperformed by the conceptually simpler Data Banzhaf in the task of detecting highly influential points, except for highly imbalanced multi-class problems.

## 1 Definitions and notation

We define **data valuation** as the task of assigning a scalar value to training points which measures their contribution to the estimated performance of a supervised machine learning model.[1] This sets us in the framework of model-agnostic data valuation (in the sense that any model can be used, but one is needed), and we specifically focus on the class of methods based in *marginal contributions*. These define the **value** of a datum $z_i$ in the training set $T := \{z_i = (x_i, y_i) : i = 1, \ldots, n\}$ as a function of its **marginal utility**, which is the difference in performance when training with and without $z_i$, measured over a separate **valuation set** $D$. A third set $D_{\text{test}}$ for testing is held out and only used for final evaluation.

The simplest example of a valuation method based on marginal contributions is **Leave-One-Out (LOO)**, which is defined as the marginal utility of $z_i$ for the whole $T$:

$$v_{\text{loo}}(z_i) := u(T) - u(T \setminus \{z_i\}), \tag{1}$$

where the **utility** $u = u_D : 2^T \to \mathbb{R}$ is the performance of the model when trained on any $S \subseteq T$, measured on a held-out **valuation set** $D$. The standard choice for $u$ in classification is **accuracy**, while for regression one can take for example the negative empirical risk. Because for large training sets $T$ the contribution of single points will be vanishingly small, LOO is typically outperformed by methods averaging many marginal contributions to different subsets of $T$.

Drawing from the literature in cooperative game theory, the seminal paper DATA-SHAPLEY (Ghorbani & Zou, 2019) takes a weighted mean of the marginal utility for *every* subset $S \subseteq T \setminus \{z_i\}$, with weights given by the Shapley coefficients:

$$v_{\text{shap}}(z_i) := \frac{1}{n} \sum_{S \subseteq T \setminus \{z_i\}} \binom{n-1}{|S|}^{-1} [u(S \cup \{z_i\}) - u(S)]. \tag{2}$$

---

[1]The field includes a wider range of definitions of value. For an overview of goals and applications of data valuation, we refer to Sim et al. (2022) as well as to the documentation of PYDVL (TransferLab, 2022).

Because of the exponential cost of $\mathcal{O}(2^{|T|})$ evaluations of the utility (each of which entails retraining the model of interest), previous work on Shapley value (Castro et al., 2009) had proposed iterating over permutations to increase the accuracy of Monte Carlo estimates: A random permutation of the index set of $T \setminus \{z_i\}$ is sampled, and one iterates over the permutation, incrementally computing marginal utilities. The authors of Ghorbani & Zou (2019) truncate this process after the inclusion of a new point results within a certain threshold of the total utility. Their method **Truncated Monte Carlo Shapley** (TMCS) can drastically reduce computation time, thus enabling Shapley-based methods for ML applications for the first time.[2]

This spurred a series of works proposing different Monte Carlo estimators (Okhrati & Lipani, 2021), variations of the Shapley value (Kwon et al., 2021), and other game-theoretic solution concepts like the Least Core (Yan & Procaccia, 2021; Benmerzoug & de Benito Delgado, 2023) (LC in the sequel), which we include in our analysis. Several sampling strategies to reduce the variance of the Monte Carlo estimates exist (Wu et al., 2023; Covert et al., 2024), but generalizations to so-called **semi-values** are more successful. The idea is to change the weights in Equation (2) with the goal of compensating for the variance of $u(S)$, which typically increases with model capacity at low set sizes $|S|$. Beta-Shapley (Kwon & Zou, 2022) gives more weight to subsets $S$ in the lower- to mid-size range by using a Beta function for the coefficients. A simpler approach, Data Banzhaf (Wang & Jia, 2023) (DB in the sequel), defined as follows, is of particular interest to our analysis:[3]

$$v_{\text{bzf}}(z_i) := \frac{1}{2^{n-1}} \sum_{S \subseteq T \setminus \{z_i\}} [u(S \cup \{z_i\}) - u(S)]. \tag{3}$$

As mentioned, the main motivation behind DB is addressing the stochasticity in $S \mapsto u(S)$ (cf. Section 3.5 for more on randomness). Intuitively, the constant coefficients $2^{n-1}$ are the best one can do for general $u$, since given any weighting scheme, it is always possible to adversarially construct a utility with high variance at the set sizes with highest weights. The authors do indeed prove certain optimality results with respect to (wrt.) a notion of stability in value rankings. We will see in the experiments that this simple idea yields the best results in many situations.[4]

Further lines of work tackle the exponential complexity by learning a model for the utility after a few evaluations (Wang et al., 2022), or by replacing the values with a proxy model altogether Jia et al. (2021). Alternative approaches avoid the cost of game solution concepts altogether, like DATA-OOB Kwon & Zou (2023) which uses the out-of-bag error estimate of a bagging model. Reformulations of Shapley like AME Lin et al. (2022) use off-the-shelf models like LASSO to look for sparse solutions with many points being assigned zero value, since one is typically interested in the extreme situations.[5]

In this context, CS-SHAPLEY Schoch et al. (2022) (CS in the sequel) appears as a valuation method based on Shapley values exclusively designed for classification problems, and aware of the in-class and out-of-class impact of training points. The core observation motivating the work (Claim 1) is that mislabeled points can simultaneously improve overall accuracy (the utility), while being detrimental for the model's performance on the class they actually belong to. The authors propose that a better definition of value should account for this phenomenon, and introduce a novel utility Equation (4) which considers the positive or negative impact a datum has within its class and outside it.

To define CS, fix $z_i = (x_i, y_i) \in T$ and split the valuation data into $D_{y_i}$, the data with the same label as $x_i$, and its complement $D_{-y_i} := D \setminus D_{y_i}$. Analogously, $T = T_{y_i} \uplus T_{-y_i}$ is a partition into the subsets of all training data with the same and different class than $y_i$, respectively. Trained over any $S \subseteq T$ the model

---

[2]The rationale for the truncation is that as the size $k$ of the subsets $S$ grows towards $|D|$, individual points contribute less and less to the performance of a model when trained on $S$, as happens with LOO. Some estimates from stability theory show that it is reasonable to expect an upper bound on the utility of $\mathcal{O}(1/k)$. It is interesting to note that diminishing returns apply also as $k$ decreases to 0. This motivates computing (2) for a certain range of set sizes $|S| \in +[B_{\text{low}}, B_{\text{up}}]$, as proposed in Watson et al. (2023), an idea that is also behind the weighting strategy of Kwon & Zou (2022) described later.

[3]Despite the formulation with a sum over the powerset of $T \setminus \{z_i\}$, as with all game-theoretic methods, in practice one does not draw subsets $S$ from it, but iterates over permutations as described above, or uses another sampling strategy. To this avail, Wang & Jia (2023) introduced Maximum Sample Reuse, an efficient strategy with good performance, see below.

[4](Li & Yu, 2023) recently extended DB to *weighted* Banzhaf values, but we were not able to include this method in our experiments.

[5]This is by far not an exhaustive list, but we want to restrict ourselves mostly to methods we discuss.

has **in-class accuracy**:[6] $a_S(D_{\boldsymbol{y}_i}) := (\# \text{ correct predictions over } D_{\boldsymbol{y}_i})/|D|$, and **out-of-class accuracy**: $a_S(D_{-\boldsymbol{y}_i}) := (\# \text{ correct predictions over } D_{-\boldsymbol{y}_i})/|D|$. Given an arbitrary set $S_{-\boldsymbol{y}_i} \subseteq T_{-\boldsymbol{y}_i}$, the **conditional utility** conditioned on $S_{-\boldsymbol{y}_i}$ is defined for every $S_{\boldsymbol{y}_i} \subseteq T_{\boldsymbol{y}_i}$ as:

$$u(S_{\boldsymbol{y}_i}|S_{-\boldsymbol{y}_i}) := a_S(D_{\boldsymbol{y}_i})\mathrm{e}^{a_S(D_{-\boldsymbol{y}_i})}, \tag{4}$$

where we define $S := S_{\boldsymbol{y}_i} \cup S_{-\boldsymbol{y}_i}$ in order to compute the accuracies.[7] Because of this form and $a_S(D_{\boldsymbol{y}_i}) + a_S(D_{-\boldsymbol{y}_i}) \leqslant 1$, $a_S(D_{-\boldsymbol{y}_i})$ is controlled by $a_S(D_{\boldsymbol{y}_i})$ in the sense that when the latter is small, the effect of out-of-class accuracy on $u$ is negligible, cf. (Schoch et al., 2022, Figure 2). In particular, even a small in-class accuracy leads to greater utility than perfect out-of-class accuracy, cf. (Schoch et al., 2022, Property 1, p. 5). With these ingredients, Schoch et al. (2022) define the **conditional CS-Shapley value** of $\boldsymbol{z}_i$ given $S_{-\boldsymbol{y}_i}$ as:

$$\phi(\boldsymbol{z}_i|S_{-\boldsymbol{y}_i}) := \sum_{S_{\boldsymbol{y}_i} \subseteq T_{\boldsymbol{y}_i} \setminus \{\boldsymbol{z}_i\}} \binom{|T_{\boldsymbol{y}_i}| - 1}{|S_{\boldsymbol{y}_i}|}^{-1} [u(S_{\boldsymbol{y}_i} \cup \{\boldsymbol{z}_i\}|S_{-\boldsymbol{y}_i}) - u(S_{\boldsymbol{y}_i}|S_{-\boldsymbol{y}_i})]. \tag{5}$$

Finally, the **CS-Shapley value** is an average over all possible **out-of-class environments** $S_{-\boldsymbol{y}_i}$:

$$v(\boldsymbol{z}_i) := \frac{1}{2^{|T_{-\boldsymbol{y}_i}|}} \sum_{S_{-\boldsymbol{y}_i} \subseteq T_{-\boldsymbol{y}_i}} \phi(\boldsymbol{z}_i|S_{-\boldsymbol{y}_i}). \tag{6}$$

In practice, this sum is approximated in Monte Carlo fashion with a few hundred $S_{-\boldsymbol{y}_i}$. For each $S_{-\boldsymbol{y}_i}$, sampling of the $S_{\boldsymbol{y}_i} \subseteq T_{\boldsymbol{y}_i} \setminus \{\boldsymbol{z}_i\}$ is not done from the powerset as suggested by Equation (5), but following the permutation approach first proposed in Castro et al. (2009). In the implementation of Schoch et al. (2022) and ours, one permutation per set is used.

## 2  Scope of the reproduction

In this report, we set to verify the main claims of Schoch et al. (2022), incorporating two additional methods: Least Core (Yan & Procaccia, 2021) and DB (Wang & Jia, 2023), which led to some unexpected results. To the best of our knowledge, this represents the first direct comparison of DB with other methods in the tasks described in Section 3.4. This work also extends the evaluation of LC conducted in Yan & Procaccia (2021) and replicates the findings of Benmerzoug & de Benito Delgado (2023) across many more datasets and scenarios. Finally, we strive to provide more accurate error estimates and include additional metrics.

**Claim 1** *Schoch et al. (2022) Training points can be simultaneously beneficial for average accuracy, and detrimental for in-class accuracy. A valuation method accounting for this should perform better than one that does not.*

This observation is backed by (Schoch et al., 2022, Figure 1), using one point and one set. How often does it happen that the marginal global accuracy of a datum is positive, but the marginal in-class accuracy is negative, and vice versa? And, how do these situations correlate with better performance of CS wrt. other methods? We address these questions in Section 4.1.

**Claim 2** *Schoch et al. (2022) CS is generally better suited for classification problems.*

We find that it is in fact DB which outperforms all methods in the detection of highly influential points, except in the case of highly imbalanced multiple classes, although it fails to do so in noise detection. See Section 3.4 for a description of these tasks, and Sections 4.2 and 4.4 for the conclusions.

**Claim 3** *Schoch et al. (2022) Data values can be successfully transferred across classifiers, including to neural models.*

---

[6]We follow the notation of Schoch et al. (2022), but observe that the sub-index in $a_S$ is a variable. A more obvious notation would be $a(S, D_{\boldsymbol{y}_i})$.

[7]There is a problem with the original notation in the paper: In p. 4 it is stated that $S_{-\boldsymbol{y}_i}$ is the complement in $S$ of $S_{\boldsymbol{y}_i}$, which is not what is intended. Instead $S_{-\boldsymbol{y}_i}$ is an arbitrary subset of $T_{-\boldsymbol{y}_i}$.

We partially verify the claim for one scenario in Section 4.3, but observe that the signal is rather small, if at all present in many cases. Using a second target classifier, we observe an almost complete failure to transfer values. This does not correlate with any obvious characteristic of the data, like class imbalance or the prevalence of the property described in Claim 1, leaving open for a practitioner the fundamental question of what source model to use for value transfer for a particular dataset.

Additionally, we observe the following (these claims are our own):

**Claim 4** *Under full randomness of data sampling, subset sampling, and training method, CS tends to exhibit higher variance than most other methods in the point removal task.*

When we resample the datasets for each experiment run, we note that the behaviour of CS is more strongly affected by the training / valuation / test split than other methods. This is of relevance since in practice, cross-validation of values is computationally prohibitive. Nevertheless, we observed that the general trends were respected across splits.

**Claim 5** *Under similar computational budgets, DB is preferable to all other methods for the identification of highly influential points, while CS is for the detection of corrupted ones.*

We substantiate this in Section 4.2 and Section 4.4.

**Claim 6** *A modified version of the metric WAD (cf. Section 3.4) is better suited for quantitative comparison of valuation methods.*

We address this in Section 4.5.

## 3 Methodology

We run all experiments on the same datasets and models as Schoch et al. (2022), adding two new valuation methods. Details of the datasets, and parameters of the classifiers and valuation methods follow.

### 3.1 Datasets

Datasets are from OPENML (Vanschoren et al., 2013). All but COVERTYPE and MNIST-MULTI are for binary classification. Stratified sampling was used for the splits to maintain label distribution. Image datasets underwent feature extraction using RESNET-18, then dimensionality reduction using 32 PCA components.

| Dataset | Type | Features | Instances | %positive | Training | Valuation | Test |
|---|---|---|---|---|---|---|---|
| Diabetes | tabular | 8 | 768 | 65.1 | 128 | 128 | 512 |
| Click | tabular | $9^\star$ | 3000 | 95.5 | 500 | 500 | 2000 |
| Covertype | tabular | 54 | 3000 | ($\natural$) | 500 | 500 | 2000 |
| CPU$^\dagger$ | tabular | 21 | 3000 | 46.7 | 500 | 500 | 2000 |
| Phoneme | tabular | 5 | 3000 | 70.6 | 500 | 500 | 2000 |
| FMNIST$^{\dagger\dagger}$ | image | 32 | 3000 | 50 | 500 | 500 | 2000 |
| CIFAR10$^{\dagger\,\dagger\,\dagger}$ | image | 32 | 3000 | 50 | 500 | 500 | 2000 |
| MNIST-binary$^{\dagger\,\dagger\,\dagger\dagger}$ | image | 32 | 3000 | 51.9 | 500 | 500 | 2000 |
| MNIST-multi | image | 32 | 3000 | ($\natural\natural$) | 500 | 500 | 2000 |

Table 1: Datasets used. ($\star$) 11 features declared online, but 9 effective after fetching using the openml library. ($\dagger$) CPU is originally a regression dataset, binarized using the threshold 89. ($\dagger\dagger$) FMNIST is restricted to the classes "t-shirt and tops" vs "shirts". ($\dagger\dagger\dagger$) CIFAR10 is restricted to the classes "automobile" vs "truck". ($\dagger\,\dagger\,\dagger\dagger$) MNIST-binary is MNIST restricted to the classes "1" vs "7". ($\natural$) Covertype is highly imbalanced with 7 classes with frequencies 36.5% / 48.8% / 6.2% / 0.5% / 1.6% / 3% / 3.5%. ($\natural\natural$) MNIST-multi has 10 classes with almost equal frequencies.

## 3.2 Valuation methods tested

Parameters for all methods were taken as suggested in Schoch et al. (2022) or the corresponding papers. Convergence criteria for the methods were kept as consistent as possible by stopping computation after the value for every training point had been updated at least a fixed amount of times.

| Method | Convergence crit. | Utility evaluations | Parameters |
|---|---|---|---|
| LOO | NA[†] | $\|T\| + 1$ | NA |
| TMCS (Ghorbani & Zou, 2019) | MinUpdates>500 | $\mathcal{O}(K\|T\|)$ | $\varepsilon = 10^{-4}$ |
| BetaShap (Kwon & Zou, 2022) | MinUpdates>500 | $\mathcal{O}(K\|T\|)$ | $\alpha = 16, \beta = 1$ |
| CS-Shapley (Schoch et al., 2022) | MinUpdates>500 | $\mathcal{O}(RK\|T\|/2)^{††}$ | $\varepsilon = 10^{-4}, K = 1$ |
| Data Banzhaf (Wang & Jia, 2023) | MinUpdates>5000 | $K$ | $K = 5000$ samples |
| Least Core (Yan & Procaccia, 2021) | NA | $K$ | $K = 5000$ constraints |

Table 2: Methods evaluated. Convergence criteria as provided by PYDVL (TransferLab, 2022). See the text for details on each method. (†) "NA" = Not Applicable. (††) $|T|/2$ is the expected size of a set $S_{-\boldsymbol{y}_i}$ sampled from $2^T$, and hence of its complement. $K$ is the number of samples taken, i.e. of permutations for all methods, except for Least Core, where we sample from the powerset. For CS-Shapley, $R$ is the number of context samples $S_{-\boldsymbol{y}_i}$.

**Leave-One-Out:**  Baseline, no parameters. Values given by Equation (1).

**Truncated Monte Carlo Shapley, TMCS (Ghorbani & Zou, 2019):**  This was the first efficient Shapley-based method and remains one of the most effective approximations. It iterates over permutations of the index set of $T$ to reduce variance of the Monte Carlo estimate, and heuristic stopping to reduce computation. For every permutation, marginal utilities are calculated incrementally using the next index in the permutation. The process is stopped early for a permutation when the relative change in marginal utility is below a threshold $\varepsilon$. Therefore the number of utility evaluations is $\mathcal{O}(K|T|)$.

**Beta Shapley, BS (Kwon & Zou, 2022):**  A semi-value approach like TMCS, where the weights for the marginal utilities are defined using a Beta function. For the parameters we use the best values according to the paper, $\alpha = 16$ and $\beta = 1$. We use the same permutation sampling scheme as for TMCS.

**CS-Shapley, CS (Schoch et al., 2022):**  The key parameter mentioned in the paper is the number $R$ of context samples $S_{-\boldsymbol{y}_i}$, which we mimic with MinUpdates. For each one of these, one permutation of $S_{\boldsymbol{y}_i}$ is used. Additionally, a threshold is used to compare the absolute difference between value estimates and update the values. Finally, there are two minor variants of the algorithm not detailed in the paper. Their implementation does not condition on sets $S$ with $|S| < c - 1$, where $c$ is the number of classes, which for binary problems means that one never conditions on the empty set. We use the default in PYDVL which does not include this restriction, hence sometimes conditioning on $S = \emptyset$ and setting $u(\emptyset|S) = u(S)$. We tested both without observing major differences.

**Data Banzhaf, DB (Wang & Jia, 2023):**  A semi-value approach with constant weights, with the goal of counteracting the variance in stochastic utility functions. The idea is that for any particular choice of weights there will always be a utility for which they perform poorly, thus making a constant the best choice. In particular wrt. rank stability of the methods, see Figure 9.

The paper also introduces an efficient sampling technique, dubbed *Maximum Sample Reuse* (MSR), which for every sample $S \subset T$ updates all indices in the training set. This drastically reduces the amount of utility evaluations required for approximation, by a factor $|T|$ wrt. TMCS.

The idea is that $v_{\text{bzf}}(\boldsymbol{z}_i) = \mathbb{E}_{S \sim \text{Unif}(2^{T \setminus \{\boldsymbol{z}_i\}})}[u(S \cup \{\boldsymbol{z}_i\}) - u(S)]$, and by linearity of the expectation $v_{\text{bzf}}(\boldsymbol{z}_i) = \mathbb{E}[u(S \cup \{\boldsymbol{z}_i\})] - \mathbb{E}[u(S)]$, which can be approximated by sampling sets $S_k \subset T \setminus \{\boldsymbol{z}_i\}$, splitting them into

$\mathcal{S}_{\ni i} := \{S_k : \boldsymbol{z}_i \in S_k\}$, $\mathcal{S}_{\not\ni i} := \{S_k : \boldsymbol{z}_i \notin S_k\}$ and setting:

$$\hat{v}_{\mathrm{bzf}}(\boldsymbol{z}_i) := \frac{1}{|\mathcal{S}_{\ni i}|} \sum_{S \in \mathcal{S}_{\ni i}} u(S) - \frac{1}{|\mathcal{S}_{\not\ni i}|} \sum_{S \in \mathcal{S}_{\not\ni i}} u(S).$$

In our experiments we chose to stop the method after all values had been updated at 5000 times, which means the same amount of utility evaluations. This was to provide a comparison to LC. We also ran DB with permutation sampling as TMCS and Beta Shapley, but performance was comparable or worse at a much higher computational cost.

**Least-Core, LC (Yan & Procaccia, 2021):** Another game-theoretic approach. It computes values with a stability property called **Coalitional Rationality**, which ensures (in the exact case) that every subset is assigned an aggregate value at least as large as its utility.[8] Given the different nature of the algorithm, which solves a linear program and cannot use the same convergence criteria, we choose $K = 5000$ constraints for stability and run-time considerations.[9]

### 3.3 Models for value computation

Values are computed using each of the models in Table 3 with the given parameter choices.

| Model | Changed parameters |
|---|---|
| Logistic regression | `solver='liblinear'` |
| Gradient Boosting classifier | `n_estimators=40`, `min_samples_leaf=6`, `maxdepth=2` |
| K-Nearest Neighbours | `n_neighbors=5`, `weights='uniform'` |
| SVM | `kernel='rbf'` |

Table 3: Models used to compute values and changes made to the default parameters in SCIKIT-LEARN 1.2.2.

### 3.4 Tasks for the evaluation of data valuation methods

Data valuation is particularly useful for data selection, pruning and inspection in general. For this reason, the most common benchmarks are **data removal** and **noisy label detection**. We describe these and related ones here, and present the results in Section 4.

**High-value point removal.** (Section 4.2) After computing the values for all data in $T = \{\boldsymbol{z}_i : i = 1, \ldots, n\}$, the set is sorted by decreasing value. We denote by $T_{[i:]}$ the sorted sequence of points $(\boldsymbol{z}_i, \boldsymbol{z}_{i+1}, \ldots, \boldsymbol{z}_n)$ for $1 \leqslant i \leqslant n$. Now train successively $f_{T_{[i:]}}$ and compute its accuracy $a_{T_{[i:]}}(D_{\mathrm{test}})$ on the held-out test set, then plot all numbers. By using $D_{\mathrm{test}}$ one approximates the expected accuracy drop on unseen data. Because the points removed have a high value, one expects performance to drop visibly wrt. a random baseline.

**Low-value point removal.** The complementary experiment removes data in increasing order, with lowest valued points first. Here one expects performance to increase relatively to randomly removing points before training. Additionally, every real dataset will include slightly out-of-distribution points, so one should also expect an absolute increase in performance when some of the lowest valued points are removed.

---

[8]This principle guarantees that each group is compensated with at least the value it brings in terms of the specified utility. This is deemed particularly relevant when compensating multiple data providers: as a purchaser, one would seek a system of credit allocation that motivates the contribution of data. However, we do not believe LC or Shapley-based methods to be relevant in data markets for practical reasons like the scarcity of data, which translates into highly unstable and potential unfair valuations, and the concentrated distribution of values, which make noise a major issue (because only those at the extrema are separated enough to be robust against randomness in the utility, cf. Figure 10).

[9]A natural choice would be $K = 125000$ constraints, in order to have the same order of magnitude in the number of utility evaluations to Beta Shapley and TMCS (accounting for truncation of permutations at around half length). However, the solvers often failed to converge and we had to reduce the number of constraints, settling in the end for 5000. Given that LC was not the focus of this reproduction, we decided to postpone investigation of this issue.

**Value transfer.** (Section 4.3) This experiment explores the extent to which data values computed with one (cheap) model can be transferred to another (potentially more complex) one. Different classifiers are used as a source to calculate data values. These values are then used in the point removal tasks described above, but using a different (target) model for evaluation of the accuracies $a_{T[i:]}$. A multi-layer perceptron is added for evaluation as well.

**Noisy label detection.** (Section 4.4) This experiment tests the ability of a method to detect mislabeled instances in the data. A fixed fraction $\alpha$ of the training data are picked at random and their labels flipped. Data values are computed, then the $\alpha$-fraction of lowest-valued points are selected, and the overlap with the subset of flipped points is computed.[10]

**Rank stability.** (Section 4.6) Following Wang & Jia (2023), we look at how stable the top $k\%$ of the values is across runs. Rank stability of a method is necessary but not sufficient for good results. Ideally one wants to identify high-value points reliably (good precision and recall) and consistently (good rank stability).

**Weighted Accuracy Drop.** (Section 4.5) While not a new task, the authors of Schoch et al. (2022) introduce the metric **Weighted Accuracy Drop** (WAD) as an aggregate way of measuring performance drop in high-value point removal with a single scalar. Given a fixed valuation method, with the notation above:

$$\text{WAD}(T) := \sum_{j=1}^{n} \frac{1}{j} \sum_{t=1}^{j} (a_{T[t:]} - a_{T[t+1:]}) \approx (\log(n) + C)a_{T[1:]} - \sum_{j=1}^{n} \frac{1}{j} a_{T[j+1:]}, \tag{7}$$

where we simplified the expression by using the telescopic nature of the inner sum, and the classical approximation of the harmonic series for large $n$, with $C > 0$ a constant. This weighted mean places more weight on training with the highest valued points, where one hopes to see the highest decrease in accuracy. Note that under the third randomness scenario, i.e. when keeping all but the inner sampling of methods constant, the value $a_{T[1:]}$ will be equal for all methods compared, making the first term a constant shift, and the second term the only effective one.[11]

Even though WAD is convenient for comparison, and it is possible to add confidence intervals as in Figure 7, we find that it has several drawbacks. First, it is effectively a weighted mean of the tail accuracies $a_{T[j+1:]}$, for which one could argue that stronger decays make more sense (maybe even clipping). More importantly, it does not take into account the heteroscedasticity of the accuracy. In Figure 2 we can see that the 95% CI tends to grow as data are removed from $T$, possibly reflecting the fact that after the few initial high values, many are very small, inducing changes in their ranking (see Figure 10 for the distribution of values).

**Variance Adjusted Relative Weighted Accuracy Drop.** For the aforementioned reason, we propose an adjustment based on the standard error. We also compare to random values at the same time step, an approach we believe to be more informative and easier to interpret. Finally, we keep the hyperbolic decay rate for a closer comparison to WAD, and to avoid an additional degree of freedom choosing decay rates.

Let the valuation methods be indexed with $k \in \{0, \dots, m\}$, where $k = 0$ is random valuation. Let $p \in \{1, \dots, n_{\text{runs}}\}$ index runs. Our proposed metric VARWAD for run $p$ and method $k > 0$ is defined as the average weighted difference to the mean performance achieved with random values:

$$\text{VarWAD}(T, p, k) := \sum_{t=1}^{n} w_k(t)(a_{T_k^p[t:]} - \overline{a}_{T_0[t:]}), \tag{8}$$

where $\overline{a}_{T_k[t:]} := \frac{1}{n_{\text{runs}}} \sum_{p=1}^{n_{\text{runs}}} a_{T_k^p[t:]}$ is the average performance of the (fixed) model when trained on the subsets of $T$ from the $t$-th index onwards, as sorted by decreasing value by method $k$ in all runs $p$. The differences are weighted by hyperbolic decay like WAD, and by standard error at each time-step across all

---

[10]This synthetic experiment is however hard to put into practical use, since the fraction $\alpha$ is of course unknown in practice.
[11]This changes however if one retrains with different seeds to compute accuracies.

runs: $w_k(t) := (1 - \tilde{s}_k(t))/t$, where $\tilde{s}_k(t) = \frac{s_k(t)}{\max_t s_k(t)}$ is the normalized standard error $s_k(t)$ across runs at time-step $t$ for method $k$. The effect of this correction is to further reduce contributions where noise is high, and to set them to 0 when it is at its maximum.

### 3.5 Randomness in value computations

There are three potential sources of randomness that affect the computation of Shapley-like values. The first two stem from the stochasticity of model performance, while the last one is due to the Monte Carlo approximations of the value:

1. For a fixed $S \subseteq T$, the utility $u(S)$ is affected by the randomness in the training algorithm mapping $S \mapsto f_S$. E.g. for SGD, different initial weights, and different sequences of minibatches $(B_j)_{j=1}^N \subseteq S$ will produce different models $f_S$.[12]

2. For a fixed, trained model $f_S$, there will be noise in the computation of the score. E.g. the accuracy $a_D(S)$ is an empirical risk estimate for some loss $l$ (e.g. 0-1 loss), and the distance to the true **generalization error** $\mathbb{E}_{X,Y}[l(f_S(X), Y)]$ will depend on the choice of valuation set $D$.

3. The set of samples $\{S_j \subseteq T : j = 1, \ldots, M\}$ used to approximate the sums Equation (5) and Equation (6).

If one is interested in evaluating the convergence properties of a certain valuation method (e.g. how good the Monte Carlo approximation is), one must focus on the third item. One freezes the dataset split, the choice of subsets $\{S_j \subseteq T : j = 1, \ldots, M\}$ and utility computations $u(S_j)$ (which is equivalent to retraining with the same sequence of batches / initialization / random seed), and computes the different values with these fixed. A new run of the experiment will keep the split, but sample the subsets $S_j$ anew and compute any new utilities $u(S_j)$, reusing the results for all valuation methods. This is the approach we follow whenever possible.

However, the above procedure can produce results which are misleading from a practitioner's point of view. If interested in the performance "in the wild", one wants to at least consider the sensitivity of the method to the variance in the estimation of generalization error, mainly in order to understand the impact that the size of $D$ has.[13] In addition, it is crucial to consider the stability of the valuation procedure wrt. noise in $u(S)$, e.g. for unstable stochastic training procedures. As discussed above, methods like Beta-Shapley and DB try to tackle this with better choices for the weights of marginal contributions.

### 3.6 Implementation details

We ran all experiments with the method implementations available in the open source library PYDVL v0.9.1 (TransferLab, 2022), on several high-cpu VMs of a cloud vendor. We initially used DVC to organize the experiment pipelines, but lock-file writing times after each combination of parameters where too long (in the several minutes each time). This motivated a switch to MLFLOW (Wilson et al., 2023). Code for all our experiments is available in (Semmler, 2024), including both setups and instructions on running them. In order to best compare methods and maximally reuse computation, we employed PYDVL's caching strategies for the utility function.

We consistently observed some discrepancies between the results in Schoch et al. (2022) and ours, namely different baseline performance of the classifiers across several experiments. We presume different pre-processing or sampling strategies to be the cause, but found the issue of no consequence for the purposes of this report.

---

[12]In our experiments this is however not the case for most models

[13]Note that it is not the magnitude of this error in itself, but the variation in value ranking as a function of the choice of $D$ that is of interest. Note also that we are assuming i.i.d. samples but of course $D$ might contain outliers, and mislabeled or corrupt data, thus distorting value computations.

## 4 Results

We repeat all experiments 20 times, keeping the data splits constant and reusing utility computations whenever possible. All plots use 95% centered percentile bootstrap confidence intervals with 10000 samples, unless otherwise stated. This approach is better suited than Normal CIs for random variables that are not normally distributed (like accuracy numbers) and for which only a few data are available. One consequence is that our confidence intervals tend to be larger than those seen in Schoch et al. (2022) and other literature.

Box plots display 1st and 3rd bootstrap quartiles, also using 10000 samples.

### 4.1 Dataset characteristics

We start with Claim 1, namely the existence of a certain type of points in datasets and the suitability of CS to address it. We look at aggregate statistics over subsets of $T$: we compute in-class accuracy $a_S(D_{\boldsymbol{y}_i}) := \big(\# \text{ correct predictions over } D_{\boldsymbol{y}_i}\big)/|D|$, and global accuracy changes for each training point, averaged over subsets $S \subseteq T$. By using constant coefficients in the averages, they coincide with Banzhaf values $v_{\text{in}}$, $v_{\text{glob}}$ for the respective within-class, and global accuracy utilities. We look at the fraction of points in each dataset that have either $v_{\text{in}} > \varepsilon$ and $v_{\text{glob}} > \varepsilon$, or $v_{\text{in}} < -\varepsilon$ and $v_{\text{glob}} > \varepsilon$, for various values of $\varepsilon > 0$. Such points are globally useful to the model but either beneficial (denoted ») , or detrimental (denoted <>) to their own class, respectively. CS is designed to give more weight to the former than to the latter and is expected to excel for datasets in which the latter abound.

In Figure 1 we see that CPU, DIABETES, FMNIST-BINARY and MNIST-BINARY contain non-negligible amounts of points of type <>, sometimes as much as roughly 40% as many as of type ». However, despite suggestive trends, there is no clear correlation between the frequency (and magnitude) of the phenomenon, and the gain in performance of CS wrt. other methods, as can be seen e.g. in Figures 2 and 8: for CPU and DIABETES, CS seems to leverage the presence of <> points, but this is no longer the case for FMNIST-BINARY, and also not exclusive of these datasets. This rather inconclusive result leaves open the question of how much of an effect the utility Equation (4) has, within the many sources of uncertainty involved.

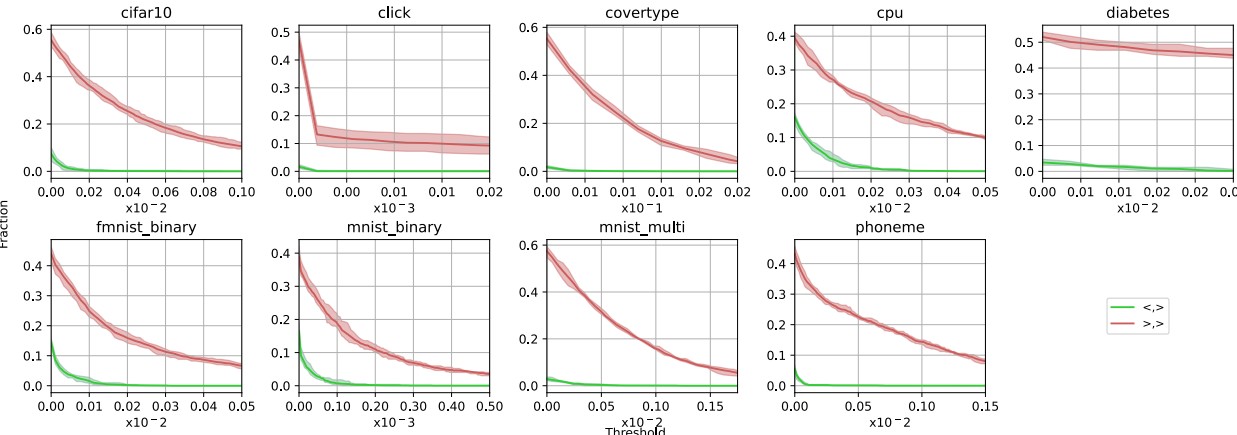

Figure 1: Prevalence of data which positively or negatively affects in-class accuracy while increasing global accuracy. The abscissa is change in average marginal accuracy. A point on the <> curve at $(x, y)$ means that a fraction $y \in [0, 1]$ of the data simultaneously induce a marginal in-class accuracy decrease and a global accuracy increase of $x$, averaged over multiple subsets $S \subseteq T$. We note that, surprisingly, the class >< (not plotted) contains anywhere from 10 to 35% of samples for most datasets at $\varepsilon = 0$. For the particular case of CLICK, almost 50% of the samples are in «, possibly indicating that the model lacks capacity

### 4.2 High-value point removal

We continue with Claim 2. In Figure 2 we are able to reproduce the experiment in Schoch et al. (2022), albeit with wider confidence intervals and some spurious differences. The exception is CLICK, where we believe

class imbalance to distort results: despite our best efforts, and stratified sampling, all methods except for DB fail at the task for this dataset (observe that more than 40% of the points are removed before a noticeable drop in accuracy). Experimentation with multiple training set splits (not depicted) shows that the general trends are respected under full randomness, indicating that for the remaining datasets, differences to Schoch et al. (2022) are mostly artifacts of the data split. We refer to A.1 for the evaluation with other models.

Qualitatively, CS is outperformed by DB (at a much lower computational budget), on 8 out of 9 datasets, as seen by the sharp initial decrease in accuracies. We recall that the number of utility evaluations used for DB is 2 orders of magnitude smaller than for the other semi-value methods. LC fails to perform in several cases with the notable exception of PHONEME, where it is unexpectedly on par with DB. For a summary evaluation with WAD and VarWAD see Section 4.5.

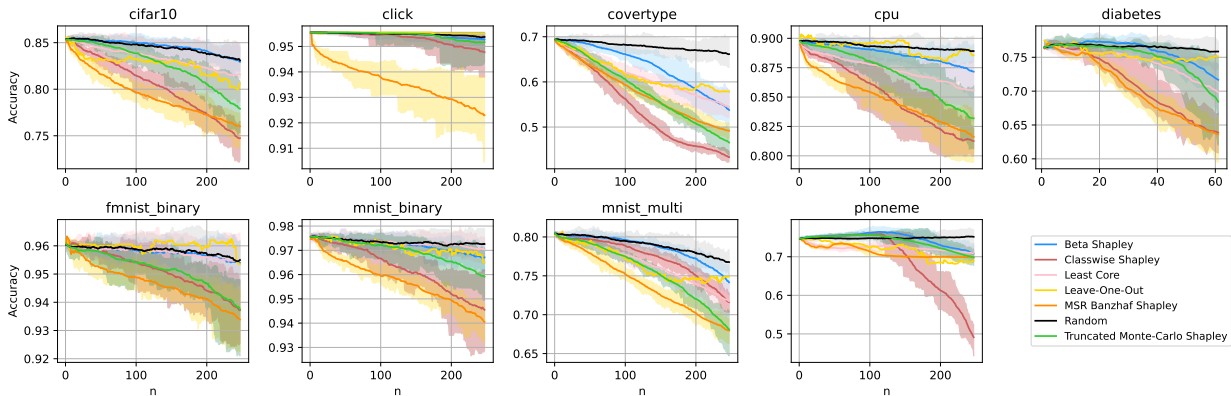

Figure 2: Accuracy drop of a logistic regression model, for values computed using logistic regression. $x$-axis is number of highest-value samples removed from the training set.

## 4.3 Value transfer

For Claim 3, we transfer values from each of the 4 models to the remaining ones, plus a neural network for evaluation, resulting in a matrix of 20 experiments. For brevity, we report only on the transfer from logistic regression to a neural network and to a gradient boosting classifier, which is the typical situation of interest: cheap value computation, expensive evaluation. For the remaining experiments, we refer to A.2.

With the neural network as target (Figure 3), we observe similar behaviour of CS as Schoch et al. (2022), again with the exception of the dataset CLICK. But in our case the addition of DB changes the landscape, since it does better in 8 out of 9 datasets. Interestingly, CS excels at the imbalanced multi-class problem. The trends are similar for all other transfers to the neural network.

For the transfer to a gradient boosting classifier (Figure 4), we experience much higher variance than Schoch et al. (2022), leading to the conclusion that *all* methods are ineffective for all but 4 datasets. In particular, we see negligible initial performance drop for 4 out of 9 datasets with most methods, invalidating the claim that the most influential samples are assigned the highest values. These, and similar mixed results for other models lead us to question the practicality of value transfer across models, despite occasional good results.

## 4.4 Noise detection

In Figure 5 we see slightly different results from those in Schoch et al. (2022): TMCS tends to perform as well as, or better than CS in most cases, while Beta Shapley does much worse. This is best seen in the AUC box plot of Figure 6, where median AUC for TMCS is typically better than for CS. Interestingly, the two cases where CS clearly wins are the imbalanced datasets CLICK and COVERTYPE, whereas it loses in the multi-class dataset MNIST-MULTI.

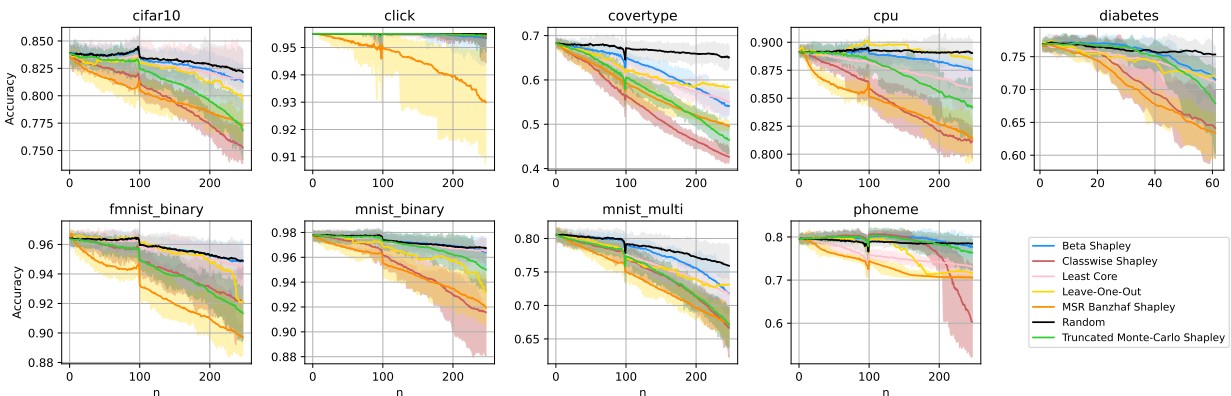

Figure 3: Accuracy drop of a fully connected neural network, for values computed using logistic regression.

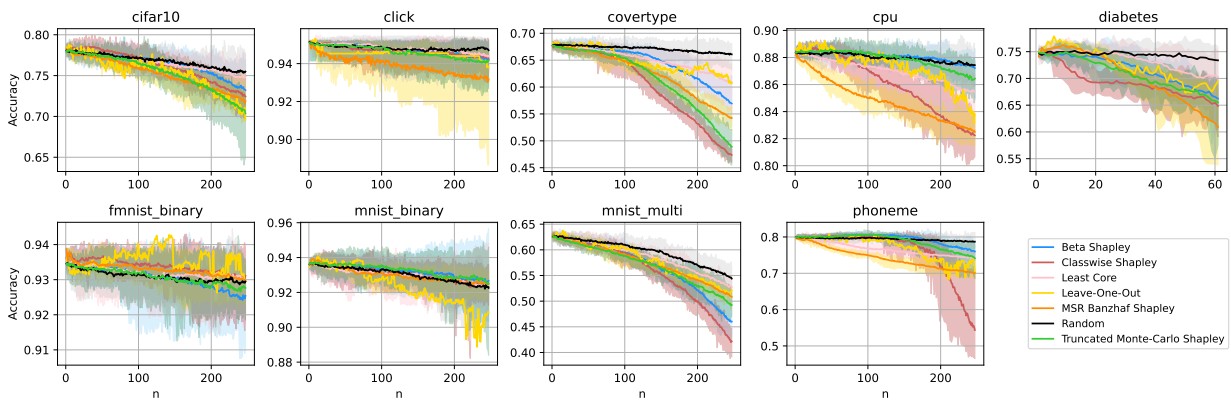

Figure 4: Accuracy drop of a gradient boosting classifier, for values computed using logistic regression.

Finally, we observe that DB performs poorly compared with the point removal task, indicating a certain insensitivity to label corruption. It is unclear why this happens, but our best guess for this phenomenon is that given a small fraction $\alpha$ of corrupted labels, they will be poorly represented among the smaller sample sizes $|S|$, and have negligible effect on the utility for the larger subsets, where they appear more often, while always being weighted with the same coefficients. Given the low rank stability seen in Figure 9, this conjecture must be taken with a grain of salt.

## 4.5 WAD and VarWAD

Following the discussion in Section 3.4, we compute WAD (with additional error bars) in Figure 7 for the same high-value removal task of Section 4.2. We then compare this to VarWAD Equation (8), which we propose as a more informative measure of aggregate performance, that better reflects the variability in performance drop.

To see this, consider dataset CLICK in Figure 2: DB exhibits better performance than other methods, albeit with high variability, and yet WAD reports similar values in Figure 7. VarWAD more accurately depicts the situation in Figure 8, setting DB ahead, while correctly reporting on its high variance. We have a similar situation in MNIST-MULTI, where WAD incorrectly ranks TMCS above DB, but VarWAD yields the intuitively correct ordering. And in CPU, where LC is relocated ahead of Beta Shapley, slightly improving the results with WAD.

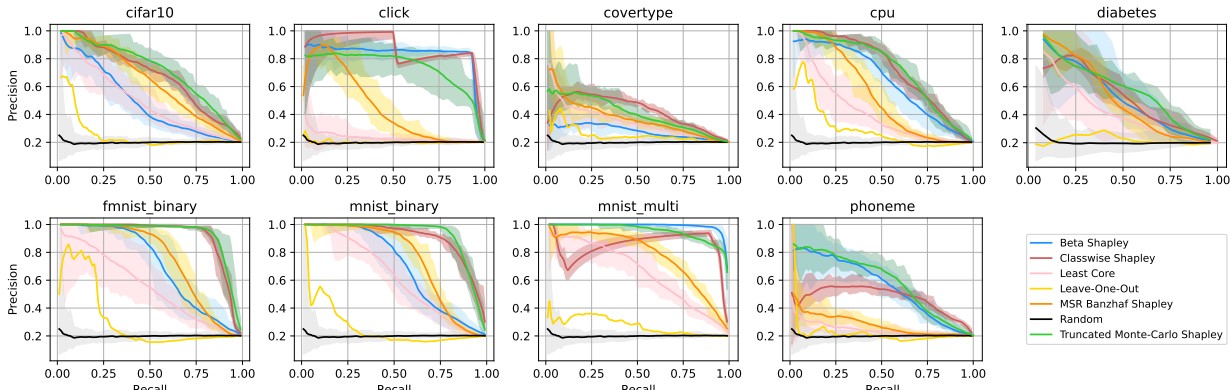

Figure 5: Precision-Recall curve for noisy label detection using logistic regression and 20% of the labels corrupted. **Precision** is the fraction of noisy samples among the top $k$ values. **Recall** is the fraction of all noisy samples identified.

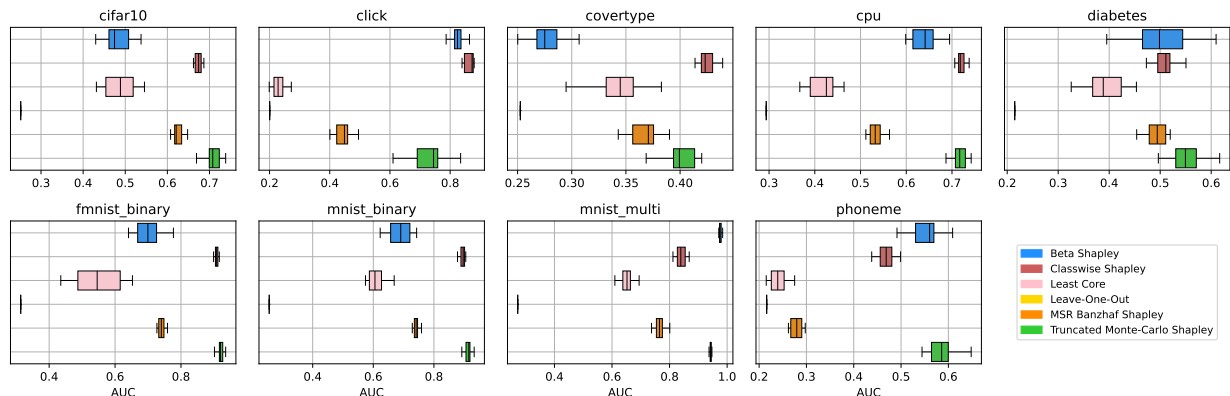

Figure 6: AUC for Figure 5, noise detection with values computed using logistic regression. Random values removed since they have the expected AUC $\approx 0.2$.

While we believe VarWAD tends to provide a more useful description of the situation, we note that these are all qualitative observations and somewhat arbitrary choices. WAD and VarWAD should thus serve as indication that it is important to design better metrics to compare these methods.

## 4.6 Rank stability

A crucial question in practice is how stable is the ranking of training points by values across runs. We look at this in Figure 9 by plotting the percentage of indices among the top $k\%$ which consistently make it to these top positions, across all runs. A value of 100% for any given $k$ means then that the top $k\%$ indices by value remain constant in every execution of the method.

The first observation is that both Beta Shapley and TMCS entirely fail to keep any fraction of points among the highest valued across runs. This instability accounts for their inability to induce stark changes in accuracy when removing the first points, as seen in Section 4.2. Next we remark how CS is more stable than DB for COVERTYPE, the highly imbalanced set where it clearly outperforms every other method. Besides this case, DB shows generally higher stability than the rest, something it was designed to achieve. In particular for CLICK, where the lack of negative cases makes training highly noisy. This is corroborated by the faster drop and wider confidence intervals seen in Figure 2.

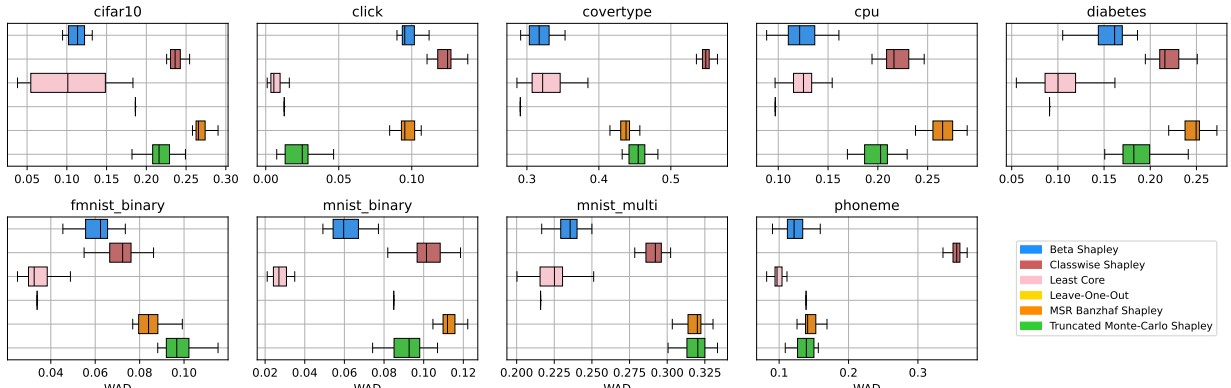

Figure 7: WAD for all datasets. Values computed with a logistic regression utility and performance drop of a logistic regression model. Because we have fixed the dataset split and reuse utility values, LOO exhibits no variance.

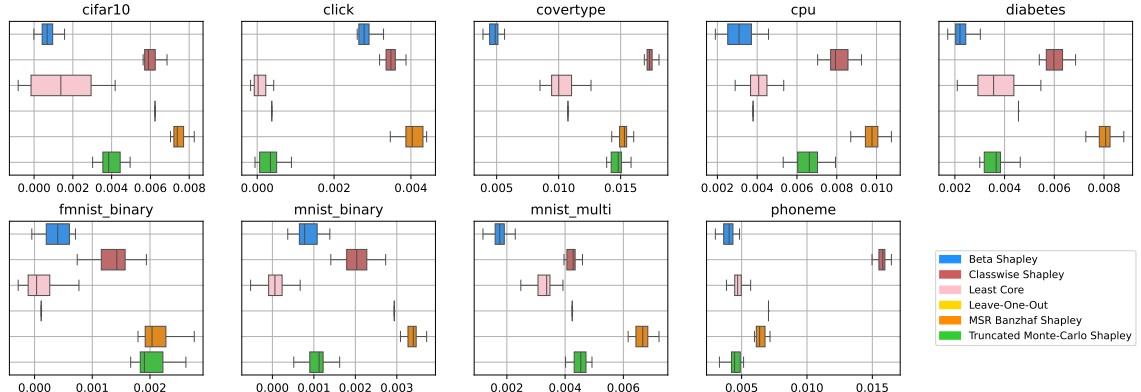

Figure 8: VarWAD for logistic regression values and point removal using logistic regression for evaluation as depicted in Figure 2. Positive numbers indicate relative improvement wrt. to random values. Because we have fixed the dataset split and reuse utility values, LOO exhibits no variance.

Generally speaking however, we observe poor stability across all datasets and methods. Given that the model used is deterministic, this means that the given computational budget is insufficient for accurate value estimation. This being a reproduction, we remained faithful to the setup in Schoch et al. (2022), but believe that a better practical understanding of sample complexity is required for applications. For deterministic methods, Maleki et al. (2014) were the first to provide Hoeffding-type bounds, which imply up to two orders of magnitude more Monte Carlo samples for an approximation with $\varepsilon = 0.01$ accuracy with probability 0.95. The most influential constant in the bound is a factor $\varepsilon^{-2}$, and the choice of this tolerance will depend on the number of training points. For Shapley values, larger datasets imply shorter intervals between values because of the *efficiency axiom*.[14] But more importantly it will also depend on the rank, because most values are concentrated around zero and with very little distance between each other, as seen in Figure 10. Further bounds in the literature prove even harder to apply in practice, e.g. those in Watson et al. (2023). All this means that it is in fact very difficult to choose the constants a priori. And the situation in the case of stochastic utilities is clearly even worse, as explained in Wang & Jia (2023).

---

[14]This axiom states that the sum of all the values must equal the total utility, and is fulfilled by Shapley values. For ML applications it has been argued that it is not essential to "distribute" a fixed amount of utility among all training points Kwon & Zou (2022), and semi-values like BS or DB dispense with it.

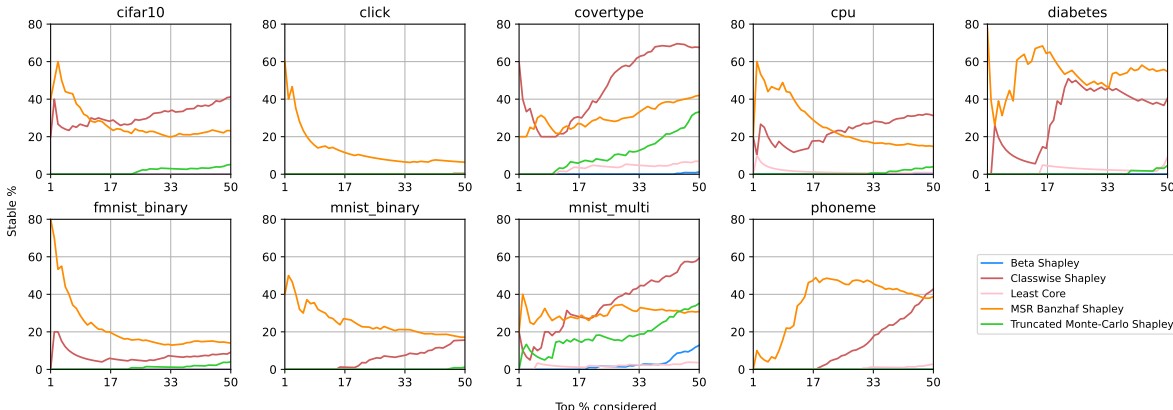

Figure 9: Rank stability of all methods. Percentage of indices ($y$-axis) that remain among the top $k\%$ ($x$-axis) across all 20 runs. Values computed using a logistic regression model. LOO is excluded because its rank stability curves are essentially flat at 1, due to the training set split (which we keep constant) being the almost unique source of randomness for LOO. Similarly we leave out random values which clearly have no stability.

## 4.7 Value decay

We conclude with an exploration of the value distributions over different datasets. Values tend to concentrate around the extrema, following a shape like the tangent function. Because of noise, mid-range values are then typically not informative.[15] Contrary to our expectation, we observe no clear correlation between the concentration of values and their rank stability or the method's performance.

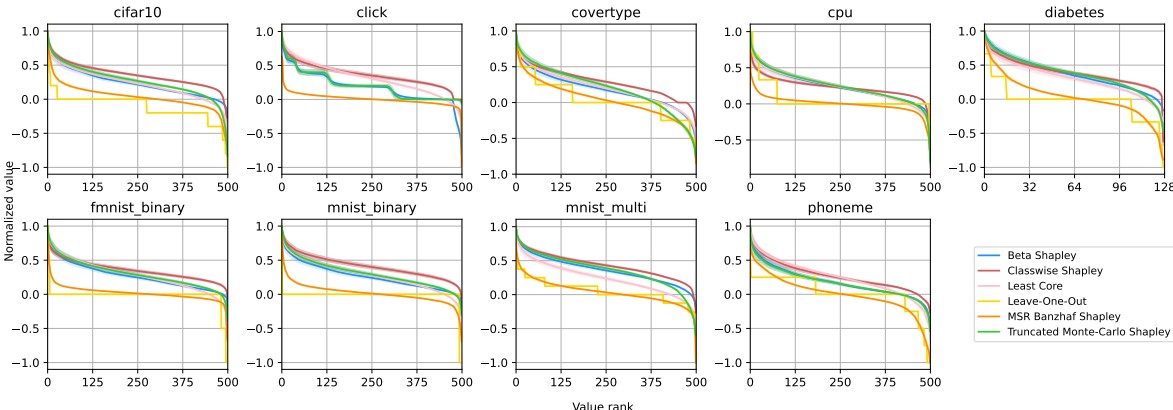

Figure 10: Value decay ($y$-axis) for all methods. $x$-axis is value rank. Values computed on a logistic regression model, sorted decreasingly and normalized for comparison. 99% $t$-student confidence interval over 20 runs.

## 5 Discussion

As is often the case with data valuation methods, the multiple sources of randomness make it difficult to draw clear-cut conclusions. We have attempted to isolate the evaluation of convergence properties of the methods, as described in Section 3.5, but the fact remains that stochasticity in $u(S)$ due to training or evaluation is the major hurdle that Shapley-based valuation methods face (besides time complexity). An interesting path

---

[15]We note in passing that the changes in curvature of the value decay function might be good spots to pick for automatic threshold selection in high-/low-value sample identification.

to explore pointed to us by a reviewer would be to use a different weighting scheme in Equation (5) and Equation (6), perhaps leading to a class-specific Data Banzhaf method incorporating the best from both.

It is alas not possible to provide clear advice to a practitioner, other than perhaps to prefer DB for high-value point identification, except for highly-imbalanced multi-class problems, where CS has proven superior. A very strong argument in favour of the former is its much greater sample efficiency when using MSR, but this technique could in principle be applied to the other methods, something that we must leave for a future more comprehensive benchmark.

We cannot unequivocally substantiate Claim 1 with our experiments and must conclude that, while definitely present, the successes of CS might be partly due to its unique utility function, partly to sampling properties and other factors. Additionally, we find that DB outperforms CS for high-value point removal with all models, except, interestingly, for COVERTYPE, in favour of the adequacy of CS for multiclass problems. The situation is reversed in noise detection, partially invalidating Claim 2 within the context of the new methods added. Attempts at transferring values from 4 models to 5 yield mixed results, with DB as a winner in many scenarios, but Claim 3 can only be accepted with caution, given the practical difficulties involved. In particular, transferred values might only useful in applications where manual inspection of the selected data is performed (which is what we recommend in any case).

Next, while DB is designed for rank stability, CS performs better in this respect for many datasets, although it must identify less important points than DB, given the worse curves in Section 4.2. This "failure" might well be due to the low sample regime, and inclusion of Maximum Sample Reuse in future benchmarks should DB is proven to be best at compensating for a worst-case noisy utility, showing that other choices of valuation method will be preferable in certain situations.

Finally, our proposed metric Equation (8) tends to capture qualitative behaviour of the methods better, but remains heuristic and arbitrary.

We conclude by remarking that the evaluations have been done at a constant computational budget (in terms of utility evaluations) when possible, but that, as explained in Section 4.6, this budget needs to be increased for most methods for reliable value estimation. The exceptions have been LC for computational reasons, and MSR Banzhaf, which we felt was only fair to the original work. Nevertheless, final performance in downstream tasks is what matters in the end and lower budgets might work just fine. All in all, marginal contribution-based data valuation methods of the sort considered in this reproduction, while still far from being fire-and-forget, automatable data selection mechanisms, remain a tool in the belt of a cautious practitioner who appreciates the value of spending time looking at their data.

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

# A    Additional experiments

## A.1    High-value point removal

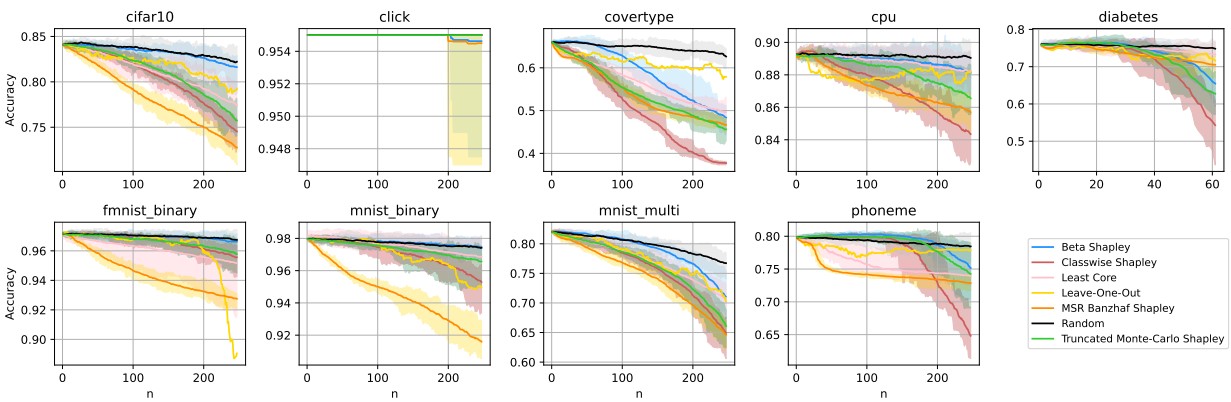

Figure 11: Accuracy drop of an SVM, with values computed on an SVM.

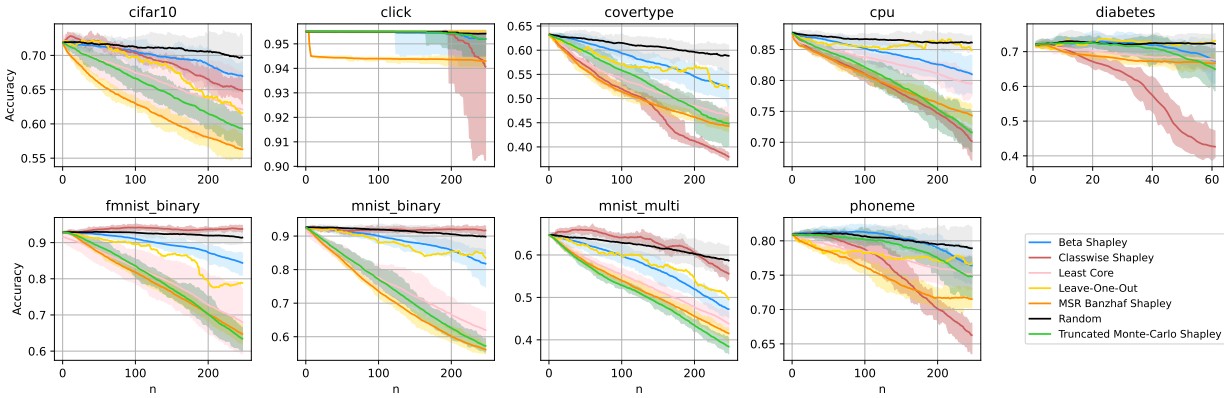

Figure 12: Accuracy drop of a KNN classifier, with values computed on a KNN classifier.

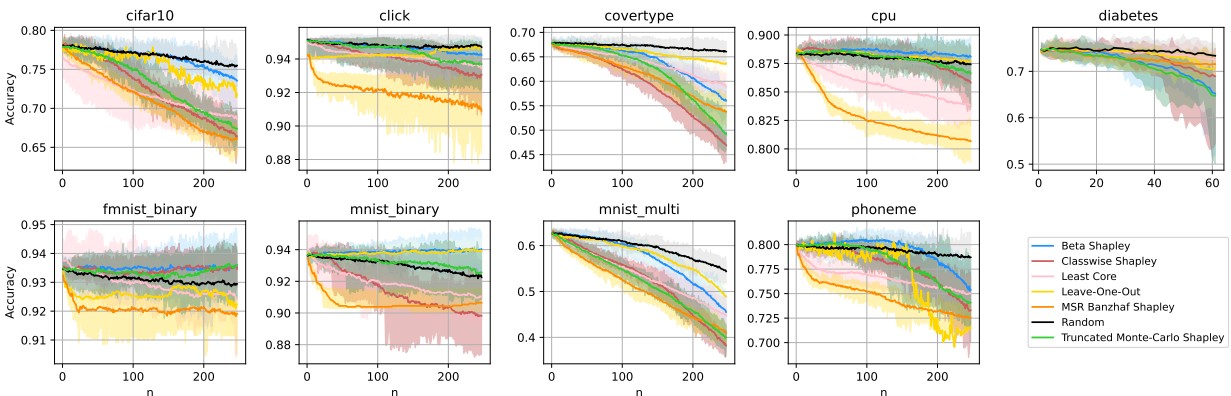

Figure 13: Accuracy drop of gradient boosting classifier, for values computed using a gradient boosting classifier.

## A.2 Value transfer

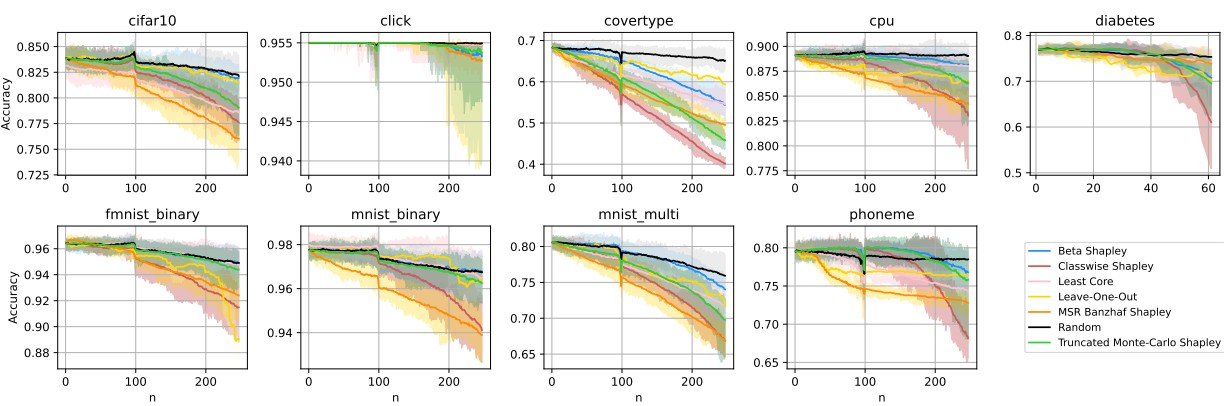

Figure 14: Accuracy drop of a fully connected neural network, for values computed using an SVM.

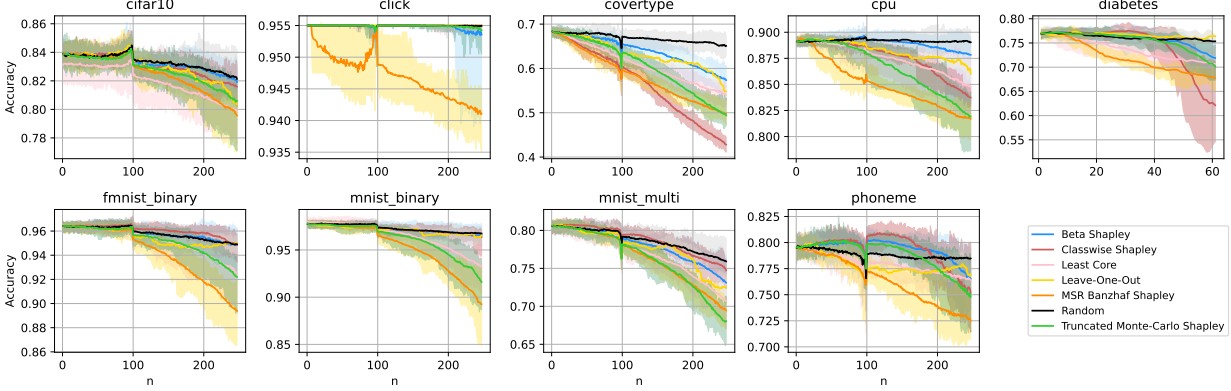

Figure 15: Accuracy drop of a fully connected neural network, for values computed using KNN.

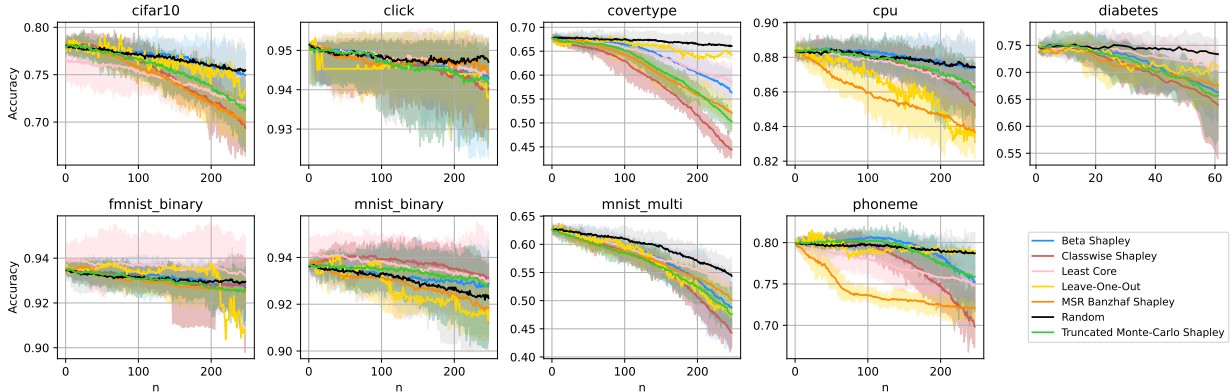

Figure 16: Accuracy drop of a gradient boosting classifier, for values computed using an SVM.

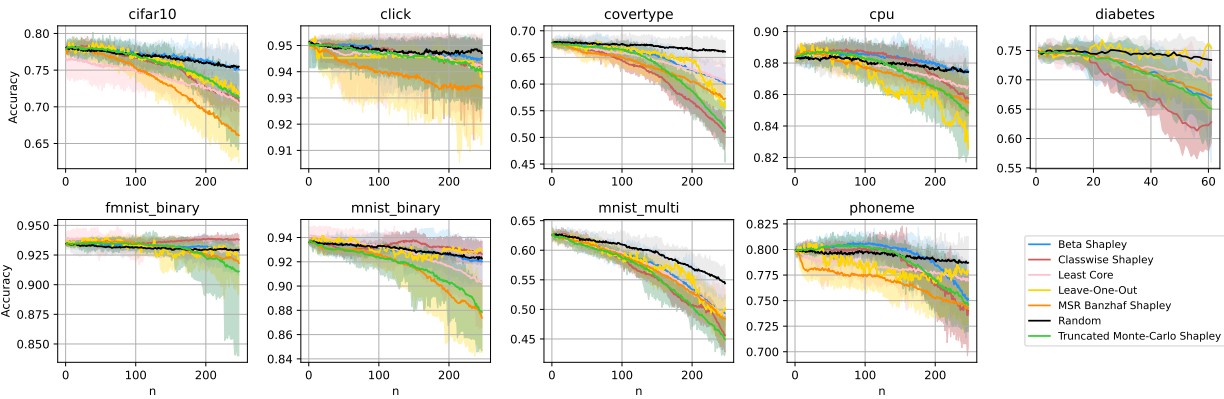

Figure 17: Accuracy drop of a gradient boosting classifier, for values computed using KNN.

