# OpenReview forum: "[Re] Classwise-Shapley values for data valuation"
_TMLR — Accepted by TMLR_

### Review · Reviewer_vPWC · 2024-03-01

**Summary Of Contributions:**

This paper is a replication study of Classwise-Shapley (CS) from Schoch et al. (2022), which examines the main proposal as well as some existing and new baselines, plus evaluation via some existing and new metrics. The paper is very clear about its setup (e.g., the models used, the amount of compute and hyperparameters used for each data valuation method). It's also clear about the main claims it evaluates, which are taken from the original work.

Overall, the paper supports certain claims from the original work and contradicts others, and this is only possible by replicating the experiments in full and adding context from missing baselines. It seems like a valuable study for helping practitioners understand the strengths and weaknesses of the CS method.

**Audience:**

Yes

**Broader Impact Concerns:**

None.

**Claims And Evidence:**

Yes

**Requested Changes:**

Several requested changes are described above.

**Strengths And Weaknesses:**

## Strengths

There are a growing number of data valuation methods and a lack of systematic comparisons between them. Furthermore, the various papers typically report only a subset of popular metrics, which creates a risk of reporting only those where the proposed method performs best. This therefore seems like a helpful reproduction study focused on CS, but which also helps us understand the comparative strengths of certain other methods (TMCS and DB). It seems to play a role a bit like the OpenDataVal library/benchmarks (which could possibly be cited), but with some methods that are not included there.

As mentioned in the summary above, I appreciated that the paper was very clear in its setup and the claims it evaluates. I found the literature review clear and helpful as well.

Among the experiments, I thought section 4.1 was particularly interesting and not sufficiently explored in the original CS work.

## Weaknesses

My feelings are mostly positive about the study, and the weaknesses I'm pointing out mostly aren't that important. But they're relatively easy changes that I believe could improve the paper.

- A nit, but in the introduction maybe "margin-based methods" should be "marginal contribution-based methods" ? Margin-based seems to connote something else.
- In footnote 2, I'm confused why Watson et al is cited for truncating the computation for large subsets, didn't Data Shapley introduce this?
- Also in the introduction, Okhrati & Lipani don't propose a variation on the Shapley value - they propose a different Monte Carlo estimator.
- On the topic of variations on the Shapley value, another recent work is "Robust Data Valuation with Weighted Banzhaf Values" (2023).
- On the topic of reducing the variance of Monte Carlo estimates, there's a recent paper that proposes an amortization trick that would work for many of the data valuation methods here: "Stochastic Amortization: A Unified Approach to Accelerate Feature and Data Attribution" (2024).
- In the middle of page 2, weigthing -> weighting
- When introducing the CS method at the bottom of page 2, can the authors explain why the original CS paper uses denominator $|D|$ in the class-specific accuracy calculations rather than the respective class counts? The only rationale I can see for this is the property $a_S(D_{y_i}) + a_S(D_{-y_i}) \leq 1$, but this doesn't seem very important (more on this next).
- Regarding the above-mentioned property, the authors write that this guarantees that for a fixed first value, an increase in the second value leads to a smaller conditional utility. The second term is put through an exponential in eq. 4, which is an increasing function, so that sounds backwards?
- It seems like the class-specific aspects of CS are orthogonal to their choice to use Shapley values, and we could just as easily define a class-specific version of Data Banzhaf (say "CDB"). Comparing CS to DB is a bit unsatisfactory, because we don't know if the performance differences are due to the class specificity or the Shapley/Banzhaf choice, and this would be best resolved by including the new CDB method. After seeing the results, I think the need for such a comparison is somewhat diminished by the inclusion of TMCS because it helps isolate the impact of class specificity in CS, but it would be nice if the authors discussed this more.
- I imagine this question could be resolved by reading Yan & Procaccia more closely, but perhaps the authors can comment on the fact that the "core" concept is not uniquely defined, or even guaranteed to exist for all cooperative games?
- I was a bit confused by the emphasis on high variance in the test set accuracy estimates. In many cases, especially those with subsampling to reduce the dataset size (e.g., MNIST), it would be easy to increase the test set sizes from Table 1 and reduce the variance.
- The VarWAD metric didn't seem very important, and the authors write that it "barely improves the ranking of WAD." What's the motivation to include it here?
- When discussing the different sources of randomness, it might be worth clarifying that most of the models here have no training stochasticity.
- In figure 2, it seems surprising that for $\epsilon = 0$ there aren't more points in the $>>$ category: wouldn't we expect the vast majority of data points to be helpful both globally and for their class? The portion of points in this category never exceeds 0.6 across all the dataset. Perhaps the authors can comment on this.
- In figure 3, the claim that all methods except DB completely fail seems like an exaggeration. Almost all the methods outperform the random baseline, so that seems like some form of success?
- None of the figures include x-axis labels. These are sometimes but not always explained in in the figure captions, and it would be much easier for readers to include them in the plots.
- The rank stability metrics in figure 10 are concerning, they seem to imply that none of the methods use nearly enough compute, including CS and DB. If this paper were proposing a new method and using these experiment settings, I could see this being grounds to reject the paper because none of the valuation scores are accurately computed; but since it's a replication paper settings from the previous works, maybe there should instead just be a stronger statement about this shortcoming in prior works? It seems to underline the need for much more efficient estimation methods.
- As an example of where the missing x-axis labels is a problem, I can't understand the x-axis in figure 11 and the relevance of 50% of the training data. It might also be helpful to clarify whether the normalization preserves negative values or forces all scores to be positive.

---

> ### Author Response · Authors · 2024-03-13
>
> We thank the reviewer for their thorough and insightful remarks (and the references!). We agree with all of them and have done our best to integrate most of them into the paper. Here are some additional clarifications:
>
>   * In footnote 2...
>
> Data Shapley introduced truncating large subsets, but Watson et al. derive bounds for small and large sets. We made this more precise in the text.
>
>   * When introducing the CS method at the bottom of page 2, can the authors explain why...
>
> Technically, this implies that $a\_{S}$ is additive over disjoint sets, $a\_{S} (D\_{y\_{i}}) + a\_{S} (D\_{- y\_{i}}) = a\_{S} (D)$, a property used in the OP's proof of Theorem 2. The purpose of this and Theorem 1 is to justify the particular structure of the utility function among all possible utilities built as a product of two continuous, monotone functions of in-class and out-of-class accuracies, which fulfill two desiderata of the authors (themselves perhaps a bit contrived, but not unreasonable).
>
>   * Regarding the above-mentioned property, the authors write that this guarantees ...
>
> Indeed, this was wrong. We fixed it in the text.
>
>   * It seems like the class-specific aspects of CS are orthogonal to their choice to use Shapley values...
>
> This is an interesting point. We did not have time to run experiments with different weighting schemes for CS, but we agree that the choice of utility is independent. We decided to add TMCS to compare to published methods, but will definitely consider variations with the CS utility in future benchmarks. We added comments to this avail.
>
>   * I imagine this question could be resolved by reading Yan & Procaccia more closely...
>
> The core can indeed be empty for some games, but relaxing the property of coalitional rationality by a positive "subsidy" $e$, one always has a non-empty $e$-core, see e.g. [here](https://pydvl.org/stable/value/the-core/). The least core is then the solution that minimizes $e$.
>
>   * I was a bit confused by the emphasis on high variance in the test set accuracy estimates...
>
> If the reviewer is referring to the previous footnote 9, we agree that the emphasis is misplaced. It is valid in general (for small or unrepresentative $D\_{\operatorname{test}}$), and one rarely has the luxury of large test sets, so cross-validation is preferable, but the comment was actually intended for the computation of the accuracies for the utilities. We removed it since we already comment on this in Section 3.5.
>
>   * The VarWAD metric didn't seem very important, and the authors write that it "barely improves the ranking of WAD.'' What's the motivation to include it here?
>
> We apologize for the confusion and have tried to make the point more clear in the text. It was intended as a caveat: focusing on the initial points will ignore later ones. For covertype, WAD takes into account later trends, that VarWAD doesn't
>
>   * In figure 2, it seems surprising that for $\varepsilon = 0$ there aren't more points in the >> category...
>
> The "missing data" belongs to unplotted classes << and ><. The latter is typically the one with most samples, with anywhere between 10 to 35% of samples across all datasets. We've added text to this avail.
>
>   * In figure 3, the claim that all methods except DB completely fail seems like an exaggeration...
>
> The statement refers only to Click, where only after dropping 40% of the data there is some change in accuracy. We've made this more precise.
>
>   * None of the figures include x-axis labels. ..
>
> This was done in a desperate attempt to reduce page count while maintaining legibility of the plots themselves. We've added them back.
>
>   * The rank stability metrics in figure 10 are concerning...
>
> We entirely agree with the assessment and have added comments in this direction in section 4.6.
>
>   * As an example of where the missing x-axis labels is a problem, I can't understand the x-axis in figure 11 and the relevance of 50% of the training data...
>
> The 50% was intended to zoom on the positive range since we have focused on high-value point removal only, but in hindsight we see that it makes more sense to add the full curve.

---

> ### Comment · Reviewer_vPWC · 2024-05-15
> **Author response?**
>
> I'm not sure if there was a mistake in how the author response was posted, but I don't see a response to my review.

---

> > ### Comment · Action_Editor_5mv9 · 2024-05-21
> > **Re: Author response?**
> >
> > Apologies for not noticing this earlier... the authors posted a response on May 14th, but the post is not visible to reviewers, only action editors and editors in chief. __Dear authors:__ please modify the "Readers" settings on your response to Reviewer vPWC so that reviewers (especially Reviewer vPWC) can see it. Thanks!

---

> > > ### Comment · Reviewer_vPWC · 2024-05-22
> > > **Author response still not visible**
> > >
> > > I'm getting notified about being late in making an official recommendation, but the authors still haven't changed their response to be visible to reviewers.

---

> > > > ### Comment · Action_Editor_5mv9 · 2024-05-22
> > > > **Re: Author response still not visible**
> > > >
> > > > I've copied the response from the authors below. Please have a look, and let them know if you have any additional concerns. Feel free to ignore the auto-reminders about being late, they come from the TMLR system automatically.
> > > >
> > > > __== (start) copied text from authors' May 14 response to Reviewer vPWC ==__
> > > >
> > > > We thank the reviewer for their thorough and insightful remarks (and the references!). We agree with all of them and have done our best to integrate most of them into the paper. Here are some additional clarifications:
> > > >
> > > > - In footnote 2...
> > > >
> > > > Data Shapley introduced truncating large subsets, but Watson et al. derive bounds for small and large sets. We made this more precise in the text.
> > > >
> > > > - When introducing the CS method at the bottom of page 2, can the authors explain why...
> > > >
> > > > Technically, this implies that $a_{S}$ is additive over disjoint sets, $a_{S} (D_{y_{i}}) + a_{S} (D_{- y_{i}}) = a_{S} (D)$, a property used in the OP's proof of Theorem 2. The purpose of this and Theorem 1 is to justify the particular structure of the utility function among all possible utilities built as a product of two continuous, monotone functions of in-class and out-of-class accuracies, which fulfill two desiderata of the authors (themselves perhaps a bit contrived, but not unreasonable).
> > > >
> > > > - Regarding the above-mentioned property, the authors write that this guarantees ...
> > > >
> > > > Indeed, this was wrong. We fixed it in the text.
> > > >
> > > > - It seems like the class-specific aspects of CS are orthogonal to their choice to use Shapley values...
> > > >
> > > > This is an interesting point. We did not have time to run experiments with different weighting schemes for CS, but we agree that the choice of utility is independent. We decided to add TMCS to compare to published methods, but will definitely consider variations with the CS utility in future benchmarks. We added comments to this avail.
> > > >
> > > > - I imagine this question could be resolved by reading Yan & Procaccia more closely...
> > > >
> > > > The core can indeed be empty for some games, but relaxing the property of coalitional rationality by a positive "subsidy" $e$, one always has a non-empty $e$-core, see e.g. here. The least core is then the solution that minimizes $e$.
> > > >
> > > > - I was a bit confused by the emphasis on high variance in the test set accuracy estimates...
> > > >
> > > > If the reviewer is referring to the previous footnote 9, we agree that the emphasis is misplaced. It is valid in general (for small or unrepresentative $D_{\operatorname{test}}$), and one rarely has the luxury of large test sets, so cross-validation is preferable, but the comment was actually intended for the computation of the accuracies for the utilities. We removed it since we already comment on this in Section 3.5.
> > > >
> > > > - The VarWAD metric didn't seem very important, and the authors write that it "barely improves the ranking of WAD.'' What's the motivation to include it here?
> > > >
> > > > We apologize for the confusion and have tried to make the point more clear in the text. It was intended as a caveat: focusing on the initial points will ignore later ones. For covertype, WAD takes into account later trends, that VarWAD doesn't
> > > >
> > > > - In figure 2, it seems surprising that for $\varepsilon = 0$ there aren't more points in the >> category...
> > > >
> > > > The "missing data" belongs to unplotted classes << and ><. The latter is typically the one with most samples, with anywhere between 10 to 35% of samples across all datasets. We've added text to this avail.
> > > >
> > > > - In figure 3, the claim that all methods except DB completely fail seems like an exaggeration...
> > > >
> > > > The statement refers only to Click, where only after dropping 40% of the data there is some change in accuracy. We've made this more precise.
> > > >
> > > > - None of the figures include x-axis labels. ..
> > > >
> > > > This was done in a desperate attempt to reduce page count while maintaining legibility of the plots themselves. We've added them back.
> > > >
> > > > - The rank stability metrics in figure 10 are concerning...
> > > >
> > > > We entirely agree with the assessment and have added comments in this direction in section 4.6.
> > > >
> > > > - As an example of where the missing x-axis labels is a problem, I can't understand the x-axis in figure 11 and the relevance of 50% of the training data...
> > > >
> > > > The 50% was intended to zoom on the positive range since we have focused on high-value point removal only, but in hindsight we see that it makes more sense to add the full curve.
> > > >
> > > > __== (end) copied text from authors' May 14 response to Reviewer vPWC ==__
> > > >
> > > > Best regards,
> > > >
> > > > Action Editor

---

> > > ### Author Response · Authors · 2024-05-22
> > >
> > > Sorry about that. We changed the visibility to everyone.

---

> > > > ### Comment · Reviewer_vPWC · 2024-05-22
> > > > **Thanks**
> > > >
> > > > Thanks for making the review visible. The changes the authors made to the text all sound reasonable to me, the paper seems improved by the various revisions.

---

### Review · Reviewer_Wbud · 2024-04-15

**Summary Of Contributions:**

This submission is a reproducibility study of CS-Shapley, a data valuation method proposed by Schoch et al. (2022). The experiments of Schoch et al. (2022) are repeated with two additional methods compared, Least Core and Data Banzhaf, and two additional metrics, rank stability and variance-adjusted weighted accuracy drop. The experimental findings of Schoch et al. (2022) are updated in light of the additional comparisons. In particular, while CS-Shapley remains preferred for the task of corrupted label detection, it is outperformed by Data Banzhaf in identifying highly influential points. Positive results on transferring data values from one model to another are also tempered.

**Audience:**

Yes

**Claims And Evidence:**

Yes

**Requested Changes:**

In addition to addressing the weaknesses above, which I think is necessary and straightforward, I think an answer to the following would strengthen the work:
- Since Least Core is a different approach with a different measure of complexity, I wonder about the sensitivity to the somewhat arbitrary choice of $K = 5000$ constraints.

**Strengths And Weaknesses:**

### Strengths
1. I think that reproducibility work (and more general retrospective work) is inherently valuable because the ML community still has a great need for it, despite the existence of venues that now accept this kind of work.
1. I think that this particular study is thoughtfully done.
1. The additions of Data Banzhaf, Least Core, and the two metrics are a useful update.
1. Section 1 also has value in providing a survey of "marginal contribution-based methods" for data valuation.

### Weaknesses
1. As a flip side to the last point under Strengths, it is not clear what is the motivation for focusing the study on CS-Shapley, as one method among many.
1. I found many phrases, etc. to be unclear:
    1. Footnote 2, "diminishing returns apply also as it decreases to 0": What is "it" here? Still the subset size $k$?
    1. Top of page 3, "changes in $a_S(D_{-y_i})$ are controlled by changes in $a_S(D_{y_i})$, and the greater in-class accuracy is, the smaller the effect of the other is on $u$": I do not understand what is meant by one "controlling" the other. I also wonder whether "smaller" should be "larger" since the multiplier $a_S(D_{y_i})$ (the in-class accuracy) is larger.
    1. Table 2: What are $K$ and $S$? I think that $K$ is the number of permutations, except for Least Core where it is the number of constraints/subsets?
    1. Footnote 8, "scarcity of data, and the concentrated distribution of values which make noise a major issue": Could the authors please elaborate? Is the second point related to Figure 10?
    1. Below eq. (8), "corrected by standard error": What correction?
    1. Section 3.5 below the list: Does "choice of subsets" refer to the $S_j$ in item 3? Later, does "reusing them for all valuation methods" mean reusing utilities $u(S_j)$ computed for previous subsets?
    1. Section 3.6: What is GCP?
    1. Beginning of Section 4, "bootstrap confidence intervals with 10000 samples": Are these 10000 resamples of the 20 runs?
    1. Section 4.1: It would be good to show equations or refer to previous equations to define in-class and global accuracy and the aggregate statistics of these.
    1. Middle of page 9": What is OP?
    1. "a factor $\epsilon^{-2}$, and this choice ... the efficiency axiom": What is "this choice," the choice of tolerance $\epsilon$? What is the efficiency axiom?
1. Possible corrections:
    1. Bottom of page 2, "$T = T_{y_i} \uplus T_{-y_i}$ is a partition into the subsets of all data": Should "all data" be "training data"?
    1. Eq. (5): Should $n - 1$ be $|T_{y_i}| - 1$ instead? Later on in "powerset as suggested by Equation (6)," I think the reference should be to (5).
    1. Eq. (7): I do not see where the $n (n+1) / 2$ factor comes from. Should it be $\sum_{j=1}^n 1/j$ instead?
    1. Section 3.6: The link to code is missing.
    1. Figure 1 caption, "$y$% of the data": Is $y$ really a percentage, or a fraction in $[0, 1]$? At $\epsilon = 0$, should the four fractions $<>$, $>>$, $><$, $<<$ (latter two not shown) add up to 1?

### Minor comments
- Bottom of page 3, "Additionally, we observe the following": I think these are the authors' claims and not Schoch et al.'s. This could be clarified.
- Section 3.5, item 2 in the list: Would it be possible to stick to the $a_S(D)$ notation instead of introducting $\hat{\mathrm{Err}}$?

---

> ### Author Response · Authors · 2024-05-05
>
> We would like to thank the reviewer for their thoroughness, and for the very
> helpful comments. Not only has the text improved with all the changes, but we
> also found and fixed a big mistake in our varwad metric, which we also
> simplified.
>
> ## Weaknesses
>
> 1. *(…) motivation for focusing the study on CS-Shapley*
>
>    We are motivated by Claims 1 and 2 in the paper, namely the reported improved accuracy for classification problems and its justification. To the best of our knowledge, none of the many other existing methods claim to target this task in particular.
>
> 1. *(…) unclear:*
>    1. *Footnote 2*
>
>       Indeed. We've clarified this in the text
>    1. *Top of page 3*
>       There was indeed a confusion and we've fixed it. What we should've written is that out-of-class accuracy is negligible when in-class is small. Hopefully it's clearer now.
>
>    1. *Table 2*
>
>       Correct. We've clarified this in the text.
>    1. *Footnote 8*
>
>       The reasoning is that for valuation to be useful in a data market, values need to be rank-stable. If by running the valuation again one can flip the order, data providers can't trust it to be fair, or adversarial users might try to game the system with repeated submissions, etc. The smaller the amount of data available, the worse the rank-stability becomes for every method. As to the concentration, which is indeed related to Figure 10, the idea is that for many methods, most values are so close together that minimal noise is able to shift them enough to change the rank, irrespective of the precision of the Monte Carlo approximation. This inherent sensitivity to the stochasticity of the utility is addressed by e.g. Data Banzhaf and Weighted Banzhaf, but the methods can't be perfect. For this reason, it is mostly the highest and lowest values that are of interest, were differences are large enough to be relatively robust against noise in the utility
>    1. *Below eq. (8)*
>
>       We especially thank the reviewer for this catch, that led us to realize that there had been a mixup of versions in the writing of that section, as well as a bug in our code. We have updated the text, simplified the metric removing the exponential decay, deleted appendix A.3 and updated the relevant plot in Figure 8.
>    1. *Section 3.5 below the list*
>
>       Correct. We added the notation to the text for clarity.
>    1. *What is GCP?*
>
>       Google Cloud Platform. We removed this unnecessary mention of the vendor.
>    1. *Beginning of Section 4*
>
>       Yes.
>    1. *Section 4.1:*
>
>       Done for the in-class accuracy, since there is little risk of confusion with global accuracy. The aggregate statistics coincide with Banzhaf values, as described in the text. We believe that adding a formula would introduce repetition.
>    1. *What is OP?*
>
>       We forgot to add the definition: Original Paper. Instead we replaced this by the paper reference.
>    1. *“a factor $\varepsilon^{- 2}$, and this choice … the efficiency
>       axiom”*
>
>       This axiom states that the sum of all the values must equal the total utility, and is fulfilled by Shapley values. We added Footnote 13 explaining it and adding a comment on semi-values.
>
> 1. Possible corrections:
>    1. *Bottom of page 2*
>
>       Corrected.
>    1. *Eq. (5):*
>
>       Corrected.
>    1. *Eq. (7)*
>
>       For no reason, we were adding the first $n$ natural numbers! Instead we wanted to sum the harmonic series, of course. We replaced this by the classic approximation $\approx \log (n) + C$.
>    1. *Section 3.6: The link to code is missing.*
>
>       Adding it would violate the double blind process. Instead we have made it available as supplemental material. The repository is already on github and will be made public after the review.
>    1. *Figure 1 caption,*
>
>       A fraction, correct. They add up to one.
>
> ## Minor comments
>
> * *Bottom of page 3, ‘‘Additionally, we observe the following"*
>
>   Done.
> * *Section 3.5, item 2 in the list: Would it be possible to stick to*
>
>   Changed.
>
> ## Requested Changes
>
> * *Since Least Core is a different approach (…)*
>
>   A natural choice if looking at computation time would be $\sim 150.000$ constraints, in order to have a similar number of evaluations to Beta Shapley and TMCS (taking into account early truncation of the permutations). However, the assembled problem, either the linear one (to compute the subsidy) or subsequent quadratic one (to compute a unique valuation) proved hard to solve. The solvers often failed to converge and we had to reduce the number of constraints, settling in the end for 5000. Given that LC was not the focus of this reproduction, we left it at that. We acknowledge that this merits further investigation, but believe it to be a task to be solved in the library we used, pyDVL. Another (soft) argument for leaving it at that is, we included MSR Banzhaf at the same number of utility evaluations, and it showed very good performance.

---

### Review · Reviewer_Rorp · 2024-04-22

**Summary Of Contributions:**

The authors present an evaluation of CS-Shapley, a classwise data evaluation model by Schoch et al. (2022). The authors reproduce the experiments in the paper with the inclusion of two additional models as comparisons, the Least Core (Yan & Procaccia, 2021) and Data Banzhaf (Wang & Jia, 2023). The authors mentioned that there are some discrepancies in the results of their experiments vs the results of Schoch et al. (2022), possibly due to the different pre-processing or sampling strategies. The authors evaluate the methods in several experimental settings, based on the claims mentioned in the CS-Shapley paper. After the full evaluation, the authors conclude that while CS-Shapley helps in some tasks like the detection of corrupted labels, it is outperformed by  Data Banzhaf in the task of detecting highly influential points.

**Audience:**

Yes

**Broader Impact Concerns:**

I do not have any broader impact concerns for this paper.

**Claims And Evidence:**

Yes

**Requested Changes:**

Please address the concern I have in the weakness section above.

**Strengths And Weaknesses:**

Strengths:
- Thorough evaluation of CS-Shapley with various experimental settings that are relevant for practitioners.
- Two additional recent methods are included in the experiments for comparison.
- The reproduction scopes are constructed by carefully analyzing the claims made in the CS-Shapley.
- Each experimental setting is described in detail, how the CS-Shapley performance compared with other methods.

Weaknesses:
 - The authors mentioned that there are some discrepancies in the results of their experiments vs the results of Schoch et al. (2022). The authors suspected that it was possibly due to the different pre-processing or sampling strategies. I think this discrepancy needs to be investigated in more detail, on the source of the discrepancy. If it's possible, the authors should consult the CS-Shapley's authors about the experimental setting or hyperparameter tuning, to reduce the discrepancy, so that the comparison will be fairer.
- As this paper falls into the reproducibility track, a detailed description of the experimental settings needs to be presented in the paper. This is to make sure that all the experiments are conducted in the settings appropriate to the original model and facilitate fair comparison among the models. The detailed description may include the dataset splits, hyperparameter values on each model, the training algorithms and their parameters, the random seed used, as well as other settings that are relevant to the experiments. Therefore, anyone who wants to redo the reproduction can get the same results/conclusions.

---

> ### Author Response · Authors · 2024-05-14
> **Reposting reply (for revision of May 5th)**
>
> (It seems like we made a mistake posting our reply to the review, but we addressed all comments in our version of May 5th)
> We thank the reviewer for their comments, which we try to address below:
>
> * *The authors mentioned that there are some discrepancies in the results of
>   their experiments vs the results of Schoch et al. (2022). The authors
>   suspected that it was possibly due to the different pre-processing or
>   sampling strategies. I think this discrepancy needs to be investigated in
>   more detail, on the source of the discrepancy. If it's possible, the authors
>   should consult the CS-Shapley's authors about the experimental setting or
>   hyperparameter tuning, to reduce the discrepancy, so that the comparison will
>   be fairer.*
>
>   We have had access to the original code of the paper which is available
>   online. But we believe that the differences are mostly due to the dataset
>   splits. If we randomize over these as well, the error bars grow by a big
>   margin, and by-and-large account for the discrepancies we've observed. This
>   is further confirmed by the variability of LOO when randomizing the split.
>
>   Additionally, we reproduce a very similar qualitative behaviour, e.g.
>   compared to TMCS, despite the occasional difference. All of this leads us to
>   believe that the comparison is fair, in as much as it can be under such low
>   sample regimes.
> * *As this paper falls into the reproducibility track, a detailed description
>   of the experimental settings needs to be presented in the paper. This is to
>   make sure that all the experiments are conducted in the settings appropriate
>   to the original model and facilitate fair comparison among the models. The
>   detailed description may include the dataset splits, hyperparameter values on
>   each model, the training algorithms and their parameters, the random seed
>   used, as well as other settings that are relevant to the experiments.
>   Therefore, anyone who wants to redo the reproduction can get the same
>   results/conclusions.*
>
>   We note that, besides the details of sections 3.1, 3.2, and 3.3, the whole
>   code is in the supplementary material, and will be immediately made available
>   on github after the review process. We place strong emphasis on
>   reproducibility, and this code runs with DVC (or MLflow), includes exact
>   dependencies and setup instructions, and can be run in one line.
>
>   The pipeline includes all stages, from dataset downloading, splitting and
>   preprocessing to plot generation, all documented. Additionally, every
>   parameter is collected in a single configuration file. We found it
>   impractical (and redundant) to copy all these details into the body of the
>   paper, and we hope that upon inspection of the supplemental material, the
>   reviewer will agree.

---

### Author Response · Authors · 2024-05-05
**MSR Banzhaf added**

A recent release of pyDVL has allowed us to include the sampling scheme for Data Banzhaf introduced in the paper, named Maximum Sample Reuse (MSR). It is comparable to permutation sampling in performance, at a much lower computational cost. This has prompted us to substitute MSR for permutation Data Banzhaf, thus more faithfully reproducing the original method. While this does not change the main results of the reproduction, it strongly biases our recommendation in favour of DB due to its faster run time.

---

### Decision · Action_Editor_5mv9 · 2024-05-28

**Recommendation:** Accept as is

**Comment:**

The paper is well-written, with clear objectives, methods, and take-away messages. All of the reviewers voted in favor of accepting this paper, and I agree with them. Since it is a high-quality reproducibility study that goes well beyond simply re-running the code of existing methods, I recommend this paper for a "Reproducibility Certification."

**Audience:**

The topic of model-agnostic "data valuation," in which one designs a mechanism to evaluate the impact that data points have on the quality/properties of the resulting trained model, has broad applications throughout machine learning. While there are theoretical principles to guide the design of such mechanisms, their efficacy is ultimately a very empirical matter, and solid experimental studies such as this one are sure to have value within the machine learning community.

**Claims And Evidence:**

As the reviewers have all highlighted, this paper is a reproducibility study centered around previous work by Schoch et al. (2022), and the goal of the paper is to evaluate how well the main claims of that paper hold under reproduction, as well as additional comparisons (new baselines, new metrics). The experimental setup exposition and visualization of results is of high quality, and the authors provide clear evidence for their re-evaluation of the claims made in the original classwise-Shapley paper. This it not to say that the conclusions are identical, however; the authors support certain conclusions (e.g., the effectiveness of CS for detecting corrupted labels), but other methods are shown to be more effective for different tasks.